# Sterol Composition of Sponges, Cnidarians, Arthropods, Mollusks, and Echinoderms from the Deep Northwest Atlantic: A Comparison with Shallow Coastal Gulf of Mexico

**DOI:** 10.3390/md18120598

**Published:** 2020-11-27

**Authors:** Laura Carreón-Palau, Nurgül Şen Özdemir, Christopher C. Parrish, Camilla Parzanini

**Affiliations:** 1Department of Ocean Sciences, Memorial University of Newfoundland, Marine Lab Rd., St. John’s, NL A1C 5S7, Canada; nsozdemir@bingol.edu.tr (N.Ş.Ö.); cparrish@mun.ca (C.C.P.); cparzanini@ryerson.ca (C.P.); 2Centro de Investigaciones Biológicas del Noroeste (CIBNOR), El Comitán, La Paz, Baja California Sur 23205, Mexico; 3Department of Veterinary Medicine, Vocational School of Food, Agriculture and Livestock, Bingöl University, Bingöl 12000, Turkey; 4Department of Chemistry and Biolog, Ryerson University, Toronto, ON M5B 2K3, Canada

**Keywords:** *Thenea muricata*, *Aplysina* sp., *Pseudoanthomastus agaricus*, *Montastraea cavernosa*, *Buccinum* sp., *Pasiphaea tarda*, *Phormosoma placenta*, *Echinometra lucunter*, sterols, gas chromatography, mass spectrometry

## Abstract

Triterpenoid biosynthesis is generally anaerobic in bacteria and aerobic in Eukarya. The major class of triterpenoids in bacteria, the hopanoids, is different to that in Eukarya, the lanostanoids, and their 4,4,14-demethylated derivatives, sterols. In the deep sea, the prokaryotic contribution to primary productivity has been suggested to be higher because local environmental conditions prevent classic photosynthetic processes from occurring. Sterols have been used as trophic biomarkers because primary producers have different compositions, and they are incorporated in primary consumer tissues. In the present study, we inferred food supply to deep sea, sponges, cnidarians, mollusks, crustaceans, and echinoderms from euphotic zone production which is driven by phytoplankton eukaryotic autotrophy. Sterol composition was obtained by gas chromatography and mass spectrometry. Moreover, we compared the sterol composition of three phyla (i.e., Porifera, Cnidaria, and Echinodermata) collected between a deep and cold-water region and a shallow tropical area. We hypothesized that the sterol composition of shallow tropical benthic organisms would better reflect their photoautotrophic sources independently of the taxonomy. Shallow tropical sponges and cnidarians from environments showed plant and zooxanthellae sterols in their tissues, while their deep-sea counterparts showed phytoplankton and zooplankton sterols. In contrast, echinoids, a class of echinoderms, the most complex phylum along with hemichordates and chordates (deuterostomes), did not show significant differences in their sterol profile, suggesting that cholesterol synthesis is present in deuterostomes other than chordates.

## 1. Introduction

In the deep sea (i.e., below 200 m depth), light and temperature diminish while pressure increases with depth [1]. In addition, both food quantity and quality decrease along the depth gradient [2]. However, there is evidence that the abyssal food web is supported by a flux of phytodetritus from the euphotic zone, and the abundance of some populations fluctuates interannually in concert with changes in this food supply [3].

Euphotic zone primary production is driven by eukaryotic autotrophy by phytoplankton and, to a lesser extent, by macroalgae and vascular plants from marine and terrestrial sources. In the deep sea, the prokaryotic contribution to primary productivity has been suggested to be higher because local environmental conditions prevent classic photosynthetic processes from occurring [4]. In reducing environments, for example, chemoautotrophic bacteria are responsible for converting inorganic energy sources (e.g., H_2_S) into organic products that can be utilized by the rest of the community [4]. Furthermore, methanotrophic bacteria can use methane as a carbon source to synthesis macromolecules, and they can usually be discriminated from other sources with the stable isotopes of carbon with a δ^13^C of −50‰ to −100‰ [5].

The lipid signature approach, including lipid class, fatty acid, and sterol composition has been used successfully to help understand marine trophodynamics [6,7]. Typically, energy flows from lower to higher trophic levels of food webs. This energy flow is accomplished by trophic interactions within biological communities. These communities, hence, have the potential to play a major role in the flux of organic matter and may be an important food source to vertebrate consumers, including humans. However, organic matter transfer can be difficult to elucidate. In this context, lipids are important biochemical compounds in marine food webs because they are rich in carbon and are energy-dense [8]. Lipids can also be used as biomarkers in ecological studies [9,10]. The sterol composition of lower trophic levels can provide useful biogeochemical information [11,12], leading to a better understanding of ecosystem functioning. Sterols can be used in trophic ecology as they play an important role in marine organisms. They are key constituents of animal cell membranes [13] and are present in all eukaryotic taxa [14,15]; they are also precursors of steroid hormones [16] and represent essential dietary nutrients for marine organisms [17,18]. For instance, diets with limited cholesterol content (<1%) significantly decreased the growth and survival rates of early stages in the crustacean *Panaeus monodon* [18,19].

As triterpenoid biosynthesis is generally anaerobic in bacteria and aerobic in Eukarya, the major class of triterpenoids in bacteria, the hopanoids, is different to that in Eukarya, the lanostanoids, and their 4,4,14-demethylated derivatives, sterols. There is a further bifurcation in the pathway of sterol biosynthesis in that photosynthetic organisms (e.g., algae and plants) form primarily cycloartenol as the first cyclization product, whereas nonphotosynthetic organisms (animals and fungi) form mostly lanosterol. There are exceptions to this general pattern of triterpenoid distribution, with hopanoids being found in some Eukarya (e.g., ferns and lichens) and sterols being found in certain methanotrophic bacteria and in cyanobacteria. It is possible that the two pathways are very primitive and that one or other of these pathways has, for functional reasons, been subsequently selected in different groups of organisms [20]. However, none of the bacteria that were identified to have an oxidosqualenecyclase (Osc) homolog in their genome are anaerobes, providing strong evidence that sterol synthesis is an aerobic biosynthetic pathway in bacteria as it is in eukaryotes [21].

The pathway of sterol synthesis has been thoroughly studied in vertebrates, fungi, and terrestrial plants, with cholesterol and ergosterol at the end of the synthesis pathway in vertebrates and fungi, respectively, and a higher variety of sterols in plants such as campesterol, sitosterol, and stigmasterol [22]. In contrast, knowledge of sterols in invertebrates, which comprise at least 16 phyla with important phylogeny differences (Figure 1), is lagging behind. The phylum Porifera is the simplest metazoan alive, and poriferans or sponges have one germ layer and lack true tissue organization. In the phylum Cnidaria, there are two germ layers in the blastula, endoderm and ectoderm; therefore, they are diploblastic organisms. The endoderm allows them to develop true tissue, including tissue associated with the gut and associated glands. The ectoderm, on the other hand, gives rise to the epidermis, the nervous tissue, and, if present, nephridia. Species of Mollusca, Arthropoda, and Echinodermata are triploblasts which emerged within the diploblasts, with gastrulation forming three primary germ layers: the ectoderm, mesoderm, and endoderm. Among them, Mollusca and Arthropoda are protostomes, and Echinodermata, the most complex phylum along with those of Hemichordata and Chordata (which include vertebrates), are deuterostomes [23]. Cholesterol biosynthesis pathway (CBP) genes in the basal metazoans were inherited from their last common eukaryotic ancestor and evolutionarily conserved for cholesterol biosynthesis. The genomes of the basal metazoans and deuterostomes retain almost the full set of CBP genes, while Cnidaria and many protostomes have independently experienced multiple massive losses of CBP genes that might have been due the appearance of an exogenous sterol supply and the frequent perturbation of ocean oxygenation [24].

To assess the euphotic contribution from phytodetritus to deep-sea sponges, corals, gastropods, decapods, and echinoids, we used sterols, most of which are absent in prokaryotes, with the exception of modified lanosterol products [21] probably acquired via horizontal gene transfer from eukaryotes [22]. Although photoautotrophs are a rich source of sterols, some have to be altered before they suit metazoan needs, for example, the exchangeable use of cholesterol and stigmasterol [25].

Sterol biosynthesis in invertebrates does not take place or else proceeds at a slow rate; therefore, they should be provided by diet [26]. Certain phytosterols were more efficient in supporting somatic growth of the crustacean *Daphnia magna* than cholesterol (e.g., fucosterol and brassicasterol), while others were less efficient (e.g., dihydrocholesterol and lathosterol), indicating substantial differences in the assimilation efficiency and further processing within the body for these dietary sterols [27]. Therefore, phytosterols detected in consumers which have no bioconversion capacity can be used as biomarkers.

Here, we evaluated sterol composition of five phyla, Porifera, Cnidaria, Mollusca, Arthropoda, and Echinodermata, from the deep and cold northeast Atlantic Ocean to infer their food sources. We then compared three phyla, Porifera, Cnidaria [28], and Echinoidea, from Echinodermata [29] with published data to investigate if their sterol composition was affected by temperature and depth. We hypothesized that the tissues of shallow, tropical benthic organisms would better represent photoautotrophic sources, such as phytoplankton, zooxanthellae, macroalgae, and plants, than tissues in their deep-water counterparts, independently of the phylogenetic relationships (phyla) [30]. To test our hypothesis, we compared sterol profiles of Porifera, Cnidaria, and Echinoidea from two contrasting sites: the Northwest Atlantic continental slope and southwest Gulf of Mexico shallow tropical coral reefs.

## 2. Results

### 2.1. Sterol Composition of Five Benthic Phyla, Porifera, Cnidaria, Mollusca, Arthropoda, and Echinodermata, from the Cold and Deep Northeast Atlantic

Species systematics, sample size, depth, and water temperature of animals collected in the deep Northwest Atlantic off Newfoundland and in the coral reef system of Veracruz are shown in Table 1. Total sterol concentrations ranged from 1.2 ± 1 mg·g^−1^ wet weight in Porifera to 37.7 ± 12.1 mg·g^−1^ in Echinodermata. While the number of sterols ranged from five, in the arthropod *Pasiphaea. tarda,* to 17, in the sponge *Thenea muricata,* no hopanoids were detected. Porifera had the lowest proportion of cholesterol (22.8% ± 1.8%), but significantly higher proportions of phytoplankton sterols, such as 24-methylencholesterol (12.2% ± 0.4%) and episterol (11.4% ± 1.6%), the phytoplankton and red algae sterol brassicasterol (12.1% ± 0.5%), the terrestrial plant sterol *β*-sitosterol (8.9% ± 0.8%), and a zooplankton sterol 24-nordehydrocholesterol (6.1% ± 0.8%). Furthermore, from the deep, cold-water site, Porifera was the only phylum showing sterols previously reported in diatoms and brown algae such as poriferasterol (2.0% ± 0.6%) and fucosterol (3.5% ± 0.7%), and specifically in brown algae such as 4-24 dimethyl 5,7-dien-3β-ol (2.0% ± 0.6%), compared to the other phyla studied. The cnidarian *Pseudoanthomastus agaricus* had more cholesterol than sponges with 68.8% ± 0.3%; in second place was occelasterol (11.6% ± 0.7%) probably from zooplankton, followed by brassicasterol (8.0% ± 2.6%) from phytoplankton and red algae. The mollusk *Buccinum* sp. and the arthropod *P. tarda* had similar sterol composition with cholesterol at 91.7% ± 1.0% and 95.3% ± 2.0%, respectively, with low contributions of occelasterol from zooplankton at 1.9% ± 0.9% and 2.2% ± 0.9%, respectively, and a detritus contribution detected with stigmasterol at 0.6% ± 0.2% and 1.5% ± 0.6%, respectively. The echinoid *Phormosoma placenta* had cholesterol at 73.9% ± 11.0%, similar to cnidarians, with sterols previously reported in zooplankton [31] such as 24-nordehydrocholesterol (2.5% ± 1.3%) and occelasterol (7.2% ± 3.1%), as well as plant-derived sterols such as β-sitosterol (2.1% ± 1.3%), along with characteristic sterols such as 9-10 secocholesta-5(10), 6, 8 trien β-ol, an analogue of vitamin D (1.0% ± 0.2%), and lathosterol (1.5% ± 0.5%), a cholesterol precursor (Table 2).

### 2.2. Comparison of Sterol Composition of Benthic Invertebrates from the Deep, Cold Sea Northwest Atlantic and Shallow, Tropical Coastal Gulf of Mexico

As hypothesized, we found significant dissimilarity in sterol profiles between shallow tropical and deep cold-water organisms (Figure 2a). In addition, phylogeny had a significant effect on the sterol profile; however, there was a significant interaction effect between depth/temperature and phylogeny (Figure 2b). Some tropical benthic organisms had significantly more campesterol, gorgosterol, and β-sitosterol from photoautotrophic sources such as zooxanthellae and plants, independent of phylum (*t* = 8.68, Markov chain probability (*p*(MC)) = 0.001, Figure 3a). Among them, the cnidarian *Montastraea cavernosa* had significantly higher proportions of campesterol and gorgosterol from zooxanthellae, while the deep-water cnidarian *P. agaricus* had a higher proportion of cholesterol, occelasterol, and brassicasterol, thus indicating zooplankton and phytoplankton sources (*t* = 11.86, *p*(MC) = 0.001, Figure 3b). Similarly, the sponge *Aplysina* sp., from the shallow, tropical site, showed high proportions of plant related sterols β-sitosterol and stigmasterol, as well as campesterol and cholesterol. On the other hand, the deep-water sponge *T. muricata* showed significantly higher proportions of phytoplankton sterols such as 24-methylenecholesterol and episterol (*t* = 9.43, *p*(MC) = 0.001, Figure 4a). In contrast to sponges and cnidarians, echinoids did not show significant differences in their sterol profiles (*t* = 2.09, *p*(MC) = 0.082, Figure 4b). In quantitative terms, lipids in shallow, tropical benthic organisms were significantly higher than in deep cold-sea organisms. The sponge *Aplysina* sp. had 15.4 ± 5.4 mg·g^−1^ wet weight, around fivefold more lipids than *T. muricata* with 2.6 ± 1.2 mg·g^−1^, while the cnidarian *M. cavernosa* had 7.2 ± 3.6 mg·g^−1^, almost twice the content of *P. agaricus* (4.1 ± 2.2 mg·g^−1^). A similar pattern was detected in echinoids, with *Echinometra lucunter* having 12.6 ± 1.2 mg·g^−1^ of lipids and *P. placenta* having 6.0 ± 3.1 mg·g^−1^ (Table 3).

## 3. Discussion

### 3.1. Food Supply to Deep Cold Ocean Invertebrates Using Sterols, C:N Ratios, and Stable Isotopes of Carbon and Nitrogen

Sterol composition of the sponge *T. muricata* suggests apportionment from phytoplankton, brown algae, and perhaps terrestrial vascular plants (Table 2); however, its δ^13^C at −17.4‰ ± 4.9‰ is indicative of phytoplankton sources more than macroalgae or higher plants [31]. Moreover, its δ^15^N of 14.2‰ ± 0.3‰ [30] suggests a higher trophic level more related to zooplanktivore and detritivore food habits (Table 3). The zooplankton and recent detritus diet is more likely because its C:N ratio of 5.4 ± 0.3 is low compared to C:N values as high as 10 with increasing organic matter decomposition, which is more common in old detritus [32].

Similarly, the Cnidaria *P. agaricus* reflects a phytoplankton supply due the presence of 24-methylenecholesterol and occelasterol characteristic of phytoplankton. They were probably trophically transferred by zooplankton, coinciding with the high proportion of cholesterol [29]. The latter is coincident with a lower trophic level indicated by its δ^15^N mean value of 11.0‰ ± 1.4‰, a wide δ^13^C value of −13.8‰ ± 4.9‰, and a C:N ratio of 6.5 ± 3.1 [30].

The mollusk *Buccinum* sp. and the arthropod *P. tarda* had the highest proportion of cholesterol, and a low or zero proportion of the phytosterol β-sitosterol. Cholesterol is probably provided by carnivore and scavenger food habits, since crustaceans with no or limited cholesterol content (<1%) have significantly decreased growth and survival rates in early stages [18]. There is consistency with the lowest C:N ratios of 4.2 ± 0.1 and 3.8 ± 0.04, respectively (Table 3), suggesting high apportionment of nitrogen from proteins characteristic of carnivores.

Lastly, the echinoid *P. placenta* had a greater variety of sterols including phytoplankton sterols such as occelasterol and zooplankton-related sterols such as 24-nordehydrocholesterol [29], a lower proportion of cholesterol, and a higher C:N ratio of 5.3 ± 0.5, suggesting that they are omnivores. Absence of hopanoids and C stable isotopes values far from metanothropic bacteria with a δ^13^C of −50‰ to −100‰ [5] suggest that the principal food supply is eukaryotic photoautotrophic in origin.

### 3.2. Comparison of Sterol Composition of Benthic Invertebrates from the Cold, Deep Northwest Atlantic and Tropical, Shallow Coastal Gulf of Mexico

Small sample sizes were not a statistical issue due to small variation and clear differences among species and sites. Permutational multivariate analysis of variance (PERMANOVA) allows multivariate comparisons of compositional data such as sterol profiles with data with or without a normal distribution of residuals. Here, 996 permutations were performed in the main test, and, in those species with small sample sizes such as sponges, 10 permutations allowed detecting significant differences (*p* = 0.001). Furthermore, between cnidarians there were 56 permutations allowing detection of significant differences (*p* = 0.001). In contrast, echinoids with 35 permutations did not show a significant difference (*p* = 0.082) because sterol profiles of echinoids from shallow and deep sites were quite similar.

Tropical, shallow benthic organisms with a lower complexity such as Porifera and Cnidaria reflected the available food supply in their sterol composition. The most obvious difference was detected in cnidarians because *M. cavernosa* reflected zooxanthellae sterols such as gorgosterol. Zooxanthellae contribution in deep, cold-water cnidarians is limited by light. In locations with clear water, the reduced light is enough to maintain hermatypic corals deeper than 50 m; for example, in the Atlantic Ocean, *Agaricia grahamae* grows at 119 m in the Bahamas [33], whereas, in the Red Sea, *Leptoseris fragilis* grows at 145 m [34]. The deepest record in the Pacific Ocean is for *Leptoseris hawaiiensis* at 165 m in Johnston Atoll [35]. However, *P. agaricus* was collected at 1027 m; thus, even a zooplankton supply revealed by 24-methylenecholesterol, occelasterol and brasicasterol is surprising. Similarly, tropical, shallow sponges had higher proportions of macroalgae and vascular plant sterols, in contrast to deep, sea sponges, also reflecting phytoplankton sterols such as 24-methylenecholesterol and episterol. These probably were trophically transferred by zooplankton, detected with nordehydrocholesterol [29]. We did not expect to see brown algae sterol, 4-24 dimethyl 5, 7-dien-3-β-ol, in deep-water sponges. Moreover, stellasterol (cholesta-7, 22*E*-dien-3β-ol), poriferasterol (24-ethylcholesta-5, 22Z-dien-3β-ol), and fucosterol (24-ethylcholesta-5,24(28) *E*-dien-3β-ol) were present in sponges from both environments, suggesting they could be synthesized by sponges [36] or their symbionts; however, they should be from fungi or other eukaryotic symbionts at this depth of 353 m since light is not available. Five dominant end products of sterol biosynthesis (cholesterol, ergosterol, 24-methyl cholesterol, 24-ethyl cholesterol, and brassicasterol) and intermediates in the formation of 24-ethyl cholesterol are major sterols in 175 species of Fungi [37].

Echinoids from the tropical, shallow site were herbivores, as their δ^13^C of −10‰ ± 0.1‰ from macroalgae and their δ^15^N of 3.3‰ ± 0.3‰ [31] suggest a low trophic level (Table 3). In contrast, the echinoids from the cold, deep site had a more depleted δ^13^C of −14.3‰ ± 0.9‰, indicating bentho-pelagic carbon sources, and higher δ^15^N ratios of 12.3‰ ± 0.3‰ [30], indicating higher tropic levels, such as those of omnivores or even carnivores (Table 3). However, echinoid sterol profiles did not show significant differences between tropical, shallow and cold, deep sites. There are two possible explanations. The first is that echinoids can select cholesterol rich macroalgae, which could explain why the sea urchin *E. lucunter* had a high contribution of red algae *Galaxaura* sp. [31], coincident with the high proportion of cholesterol in *Galaxaura* sp. [29]. The second is that echinoids can modify phytosterols via dealkylation or synthesize cholesterol similarly to vertebrates among the chordates. The presence of lathosterol, a precursor of cholesterol, in *P. placenta* suggests that echinoderms can actually synthesize cholesterol. Lathosterol has been recorded in other cold, deep-water echinoderms such as the holoturian *Cucumaria japonica* [38]. This suggests that the maintenance of the complete set of CBP genes in vertebrates [24] is also maintained in echinoids.

Lastly, as lipids are the storage reservoir of energy in animals [19], the ratio between triacylglycerols and sterols (TAG:ST) can be used as an indicator of nutritional condition in fish larvae [39] and invertebrates [40]. Tropical, shallow sponges, cnidarians, and echinoids showed a significantly higher concentration of lipids and higher TAG:ST ratios, suggesting a better nutritional condition than deep, cold-water invertebrates, even if primary productivity was slightly higher in the Newfoundland offshore with a chlorophyll concentration of 1.24 mg·m^−3^ [41] in comparison with the Parque Nacional Sistema Arrecifal Veracruzano (PNSAV) with a chlorophyll concentration of 1.0 mg·m^−3^ [42] in similar seasons. Further investigation comparing essential fatty acids would provide more information about food quality.

## 4. Materials and Methods

### 4.1. Study Area

The geographical location and depth profile of the northwest Atlantic and southwest Gulf of Mexico sites are shown for comparison in Figure 5 and Figure 6. The cold, deep study area was located in zone 3K, which represents the marine area located off the northern coasts of Newfoundland, according to the Northwest Atlantic Fisheries Organization (NAFO) division of the North Atlantic Ocean (Figure 6; more details in [43]). Mean daily bottom temperature during the sampling period at the sampling site ranged between 3.2 and 4.5 °C, with a slight decrease with depth. This region of the Northwest Atlantic is characterized by high productivity levels and is affected by seasonal blooms of large-celled phytoplankton; for instance, in 2012, chlorophyll *a* ranged from 1.24 mg·m^−3^ in November to 2.23 mg·m^−3^ in April [41], and there are strong lateral food inputs [44,45,46]. Due to its well-known productivity, the area has been heavily exploited and fished.

In contrast, the tropical shallow samples were collected in the Veracruz reef system national park (Parque Nacional Sistema Arrecifal Veracruzano; PNSAV), which is located off the State of Veracruz, Mexico, adjacent to the cities of Boca del Rio and Anton Lizardo (19°02′24″ to 19°15′27″ north (N), 96°12′01″ to 95°46′46″ west (W)). It is part of a larger coral reef system in the Caribbean and Gulf of Mexico. A group of 13 reefs is located near Veracruz and Boca del Rio, and another group of 15 reefs with larger structures is located near Antón Lizardo; the two reefs are separated by the Río Jamapa and delimited to the north by Río La Antigua and to the south by the Río Papaloapan. During the summer, under sustained southward winds, a cyclonic eddy develops off Veracruz Port, which enhances productivity in the area reaching 2.3 ± 1.1 g·m^−3^ total plankton wet biomass [47] and a chlorophyll concentration of 1 mg·m^−3^ [42].

### 4.2. Sampling

In the cold, deep site, animals belonging to five phyla were opportunistically collected within 7 days in November–December 2013, during one of the annual multispecies bottom-trawl surveys conducted by Fisheries and Oceans (DFO), Canada. Individuals were sampled onboard the CCGS Teleost, from a total of 23 tows inside a 100 km radius and a depth range of 313 to 1407 m. The gear used to collect the samples (Campelen 1800 shrimp trawl) included a 16.9 m wide net with four panels of polyethylene twine. Once on board, individuals were immediately vacuum-packed and frozen at −20 °C to minimize lipid oxidation and hydrolysis. Individuals were identified to the lowest possible taxonomic level, from direct observation and through photo-identification. A total of 34 deep-sea organisms, belonging to five species from different phyla, were weighed, and samples of 0.7 ± 0.2 g wet weight were collected from each individual and processed for lipid analysis at the Core Research Equipment and Instrument Training Aquatic Research Cluster (CREAIT-ARC) Facility at Memorial University (Table 1). The following were sampled: pieces of sponges, body walls from cnidarians, foot muscle from gastropods, dorsal abdominal muscle from crustaceans, and wall and tube feet from echinoids.

Sponges, cnidarians, and echinoids sterols from the deep, cold-water site were compared to sterols of sponges and cnidarians [28] and echinoderms [29] from shallow tropical coral reefs. Sponges and echinoderms were collected in September 2007 and coral was collected in October 2009 in the Cabezo and Blanca reefs at depths from 18 to 20 m (Figure 7).

### 4.3. Laboratory Analysis

Samples were weighed wet, and around 0.7 ± 0.2 g was immersed in 4 mL of chloroform, sealed under nitrogen gas, and stored in a freezer (−20 °C). Lipids were extracted by adding 2 mL of methanol and 1.8 mL of distilled water, mixed with a vortex, and centrifuged to separate polar and nonpolar phases. The lower phase was recovered in a lipid-cleaned tube; an aliquot was separated into lipid classes by Chromarod thin-layer chromatography (TLC) [48]. Chromarod TLC and Iatroscan flame ionization detection (FID) were used to quantify the lipid classes in the extracts. TLC/FID was performed with Chromarods SIII and an Iatroscan MK VI (Iatron Laboratories, Tokyo, Japan). Concentrations of triacylglycerols (TAG), steryl esters, and free sterols were obtained by interpolation with a calibration curve constructed with five amounts, ranging between 0.5 and 4.0 μg, of the following standards: cholesteryl stearate, triplamitin, and cholesterol (Sigma-Aldrich, Toronto, Canada). The remainder of the extract was separated by a solid-phase extraction column (Strata SI-1 Silica 55 mm, 70 A; Phenomenex, Torrance, CA, USA). Neutral lipid fractions were recovered with 6 mL of chloroform/methanol/formic acid (98:1:0.05); then, galactoglycerolipids were recovered with 6 mL of 100% acetone, and phosphoglycerolipids were recovered with 9 mL of chloroform/methanol (5:4) as a modification of [49]. Quantification of lipids, stable isotopes, and C:N molar ratios for cold, deep samples were obtained from [30], and lipids and stable isotopes of tropical shallow echinoids were obtained from [31]. Stable isotopes and C:N molar ratios were used to assign functional group and possible organic carbon source (Table 3).

Neutral lipid, galactoglycerolipid, and phosphoglycerolipid fractions were transmethyl esterified with Hilditch reagent, and then silylated with bis trimethylsilyl trifluoroacetamide (BSTFA) (Supelco: 3-2024) [49]. Sterols from all samples were recovered in hexane and analyzed in a gas chromatograph coupled to a mass spectrometer (GC–MS (EI): Agilent 6890N GC-5973 MSD (quadrupole) with a DB-5 column 30 m × 0.32 mm × 0.25 μm, Agilent Technologies, Santa Clara, CA, USA). The quadrupole was set at 150 °C, and the mass spectrophotometer ion source was set at 270 °C and 70 eV. The positive ionization voltage on the repeller pushed the positive ions into several electrostatic lenses, producing a positive ionization mode. Peaks were identified by the retention time of standards and mass spectra interpretation. Sterols were identified by reference to [20] and [50] (p. 235). Positions of peaks were useful for identification of sterols with the same molecular ion, and the list provided by [20] was used to infer position relative to cholesterol. Mass spectra were searched with Wsearch 32 software (Wsearch 2008; version 1.6 2005, Sidney, Australia) and the NIST Mass Spectral Search Program for the NIST/EPA/NIH Mass Spectral Library (Demo version 2.0 f. build Apr 1 2009, USA). Sterol identity was confirmed by mass spectrum interpretation using the molecular weight (MW) ion (M^+^) which provides information on number of carbons and double bonds [29,50]. Some important fragments for interpretation of the whole structure were (M-R)^+^, (M-R-2H)^+^, (M-H_2_O)^+^, (MR-H_2_O)^+^, (M-R-2H-H_2_O)^+^, and (M-CH_3_-H_2_O)^+^. Other fragments indicated the position of double bonds. For instance, the lathosterol–TMS ether has a MW = 458, suggesting that the molecule has 27 carbons and one double bond. We should expect a signal at *m*/*z* = 213 indicating a double bond at C_7_. For the stigmasterol–TMS ether, double bonds at C_5_ and C_22_ are shown by ions with *m*/*z* = 129 and *m*/*z* = 255 [29] (p. 15), [50] (p. 235). Finally, three standards were injected to build calibration curves for cholesterol, campesterol, and β-sitosterol (Sigma-Aldrich). Areas under the chromatographic peak (Figure 8) were integrated with Wsearch 32 software (Wsearch 2008; version 1.6 2005), and quantification was performed by interpolation of a calibration curve [29] (pp. 17–19). No peaks were detected in the galactoglycerolipid or phosphoglycerolipid fractions; therefore, all results refer to the neutral lipid fraction.

### 4.4. Data Analysis

Species differences of individual (univariate) sterols were tested with one-way ANOVA using the Fisher statistic (F). Residual analyses were used to test assumptions of normality and equal variance. The effect of tropical shallow vs. cold deep waters on sponge, cnidarian, and echinoderm sterols was tested with multivariate analyses of sterol profile (>2% in a least one sample) using PRIMER 6.1.16 software and PERMANOVA + 1.0.6 (PRIMER-E, Plymouth, UK). Nonmetric multidimensional scaling was conducted on the basis of the Bray–Curtis similarity coefficient.

No transformation was used to avoid artificial weighting of sterols that gave only trace contributions to their respective profiles. Similarities among species and depth were investigated using the similarity percentages function (SIMPER), and statistical differences were tested with two-way analysis, for depth and phyla, using permutational multivariate analysis of variance (PERMANOVA), which allows multivariate comparisons with data with or without a normal distribution of residuals. The method of permutation of residuals under a reduced model was used, and significant differences were tested with Markov chain probability *p*(MC) values because the sample size was *n* < 30.

## 5. Conclusions

In the deep cold sites, gastropods and arthropods had cholesterol as their main sterol. Other extraction systems such as acetic acid and methanol could facilitate finding bioactive sterol metabolites for use against diabetes and oxidative stress-induced inflammatory diseases [51]. Sponges and cnidarians from shallow, tropical environments had plants such as mangrove or seagrass and zooxanthellae sterols, respectively, while cold, deep sponges had 24-ethyl sterols probably synthesized themselves [36], and cnidarians had phytoplankton, zooplankton, and probably fungal sterols. In contrast, echinoids, belonging to Echinodermata, the most complex phylum along with hemichordates and chordates (deuterostomes), did not show significant differences in their sterol profiles, suggesting that cholesterol synthesis is present in deuterostomes other than chordates. Tropical, shallow sponges, cnidarians, and echinoids showed a significantly higher concentration of lipids, as well as higher TAG:ST ratios, suggesting a better nutritional condition than deep, cold invertebrates. However, other echinoderm groups should be studied such as holothurians which did not have cholesterol as the main sterol, but rather diatomsterol, probably because supercritical fluid extraction was used [38]. Sex was the main factor affecting sterol amount in the gonads (intact, spawned) and gametes of echinoderm *A. dufresnii,* with differences in their total concentration from intact to spawned states explained by gamete concentrations [52]. As gonads are a good functional food for diabetes mellitus treatment in mice [53], sex and gonad composition of echinoderms should be recorded in future surveys.

## Figures and Tables

**Figure 1 marinedrugs-18-00598-f001:**
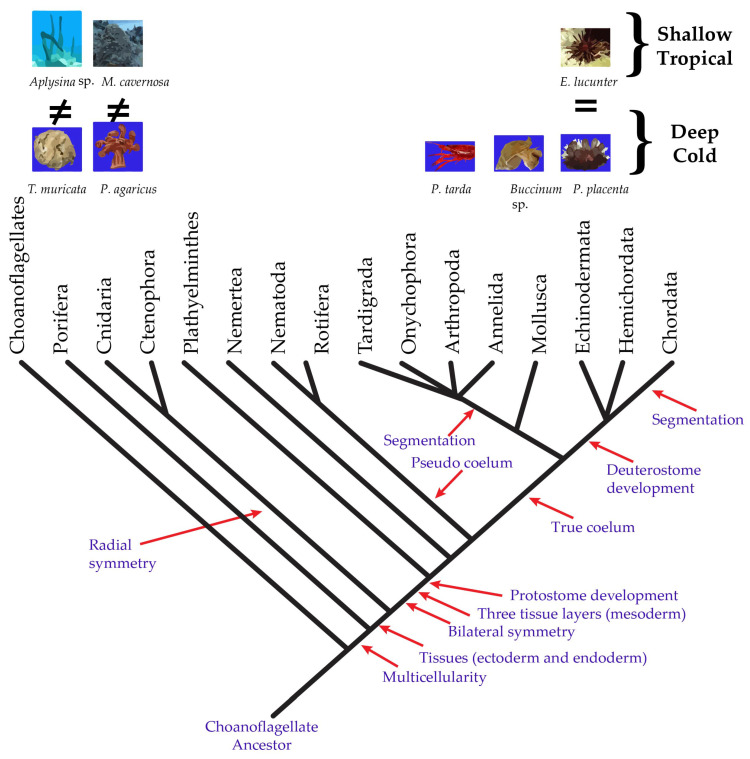
Phylogenetic position of Porifera, Cnidaria, Mollusca, Arthropoda, and Echinodermata in the tree of life of the kingdom Animalia in contrasting geographical zones.

**Figure 2 marinedrugs-18-00598-f002:**
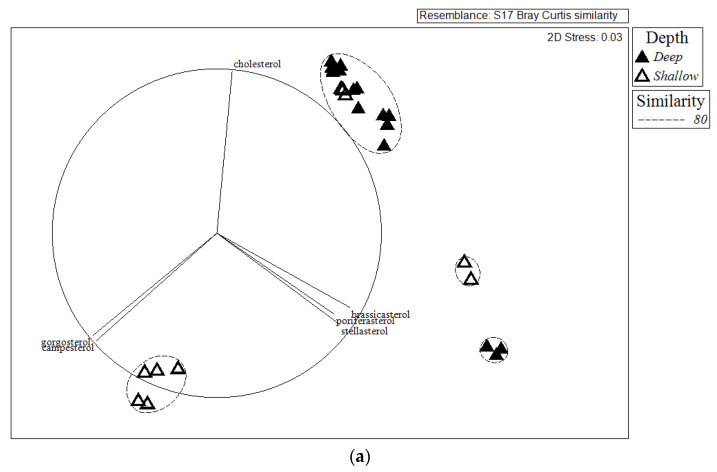
Scatter plot of nonmetric multidimensional scaling (nMDS) using Bray–Curtis similarity matrix for sterols (expressed as percentage of total sterols) of sponges *T. muricata* and *Aplysina* sp., cnidarians *A. agaricus* and *M. cavernosa*, the mollusk *Buccinum* sp., the arthropod *P. tarda*, and echinoids *P. placenta* and *E. lucunter* from deep, cold (filled symbols) and shallow, tropical (open symbols) waters. Axis scales are arbitrary in nMDS. (**a**) Sterol composition depending on depth (F_1,26_ = 75.22, *p*(MC) = 0.001), and (**b**) interaction among depth and phyla (F_2,26_ = 80.20, *p*(MC) = 0.001). Only variables with Pearson’s correlations with MDS 1 and MDS 2 >0.86 are plotted. Contours grouped with 80% similarity on the basis of hierarchical cluster analysis.

**Figure 3 marinedrugs-18-00598-f003:**
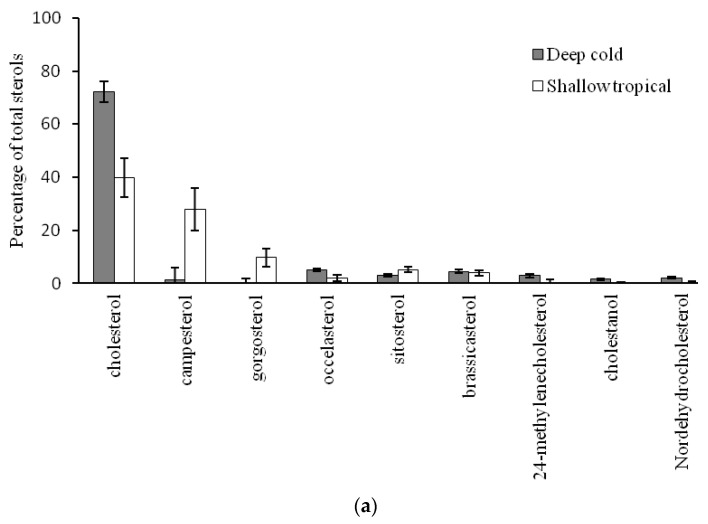
Average contribution of sterols primarily providing the discrimination between (**a**) deep, cold and shallow, tropical environments considering the three phyla studied. Sterols showed a significant similarity percentages (SIMPER) dissimilarity of 48.76% (*t* = 8.67 and *p*(MC) = 0.001). (**b**) Cnidaria *P. agaricus* (deep, cold) and *M. cavernosa* (shallow, tropical) with a significant dissimilarity of 78.76% (*t* = 11.86 and *p*(MC) = 0.001). Error bars denote 95% of the confidence interval.

**Figure 4 marinedrugs-18-00598-f004:**
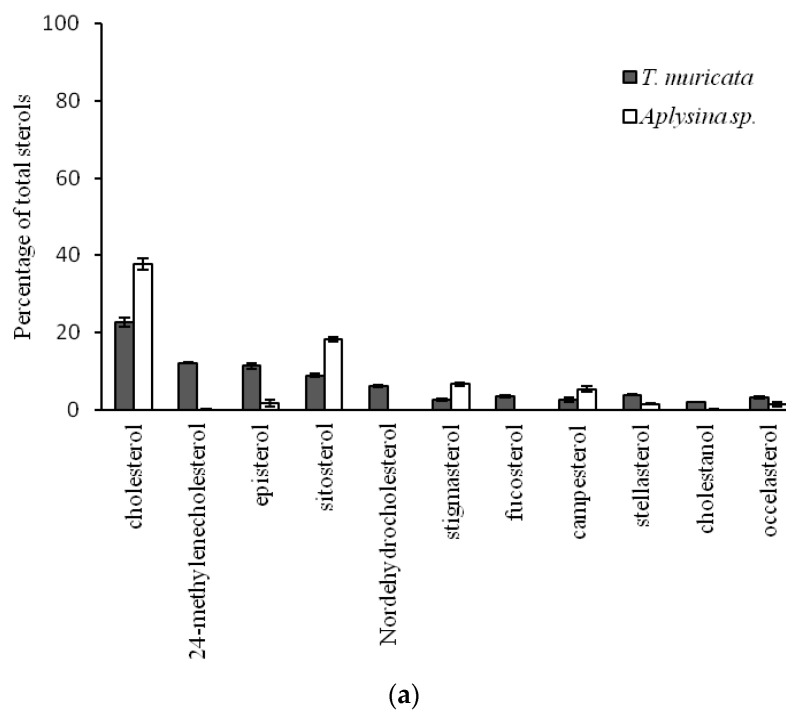
Average contribution of sterols primarily providing the discrimination between deep, cold and shallow, tropical waters. Error bars denote 95% of the confidence interval of (**a**) Porifera of factor phyla *T. muricata* (deep, cold) and *Aplysina* sp. (shallow, tropical) with a significant dissimilarity of 39.30% (*t* = 9.43 and *p*(MC) = 0.001), and (**b**) echinoids of factor phyla *P. placenta* (deep, cold) and *E. lucunter* (shallow, tropical) with no significant dissimilarity of 15.99% (*t* = 2.09 and *p*(MC) = 0.082).

**Figure 5 marinedrugs-18-00598-f005:**
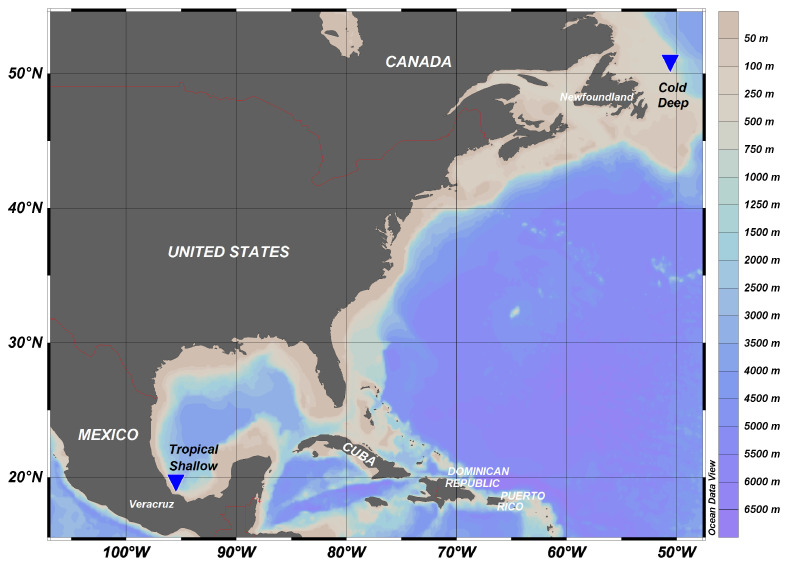
Map of sampling sites off the northeastern coast of the Province of Newfoundland and Labrador, Canada, in the Northwest Atlantic (cold, deep) and off the coast of Veracruz, Mexico in the southwest Gulf of Mexico (tropical, shallow).

**Figure 6 marinedrugs-18-00598-f006:**
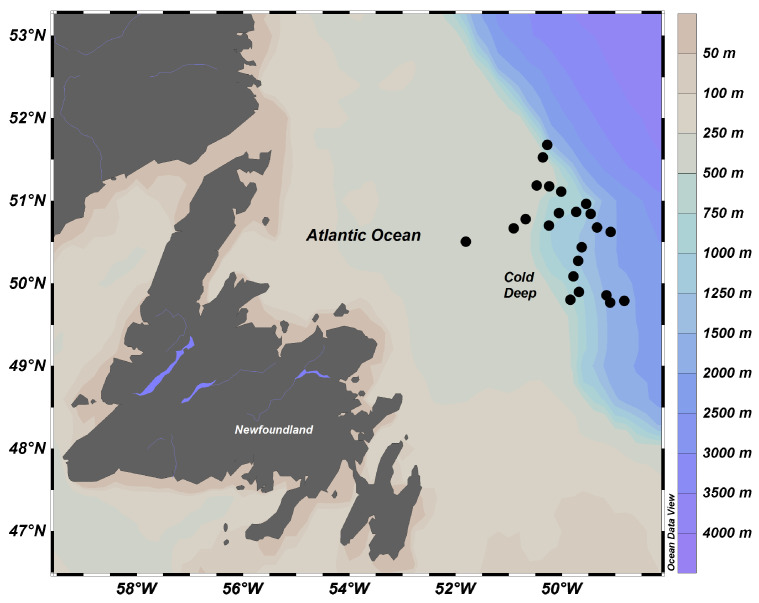
Map of sampling sites off the northeastern coast of the Canadian province of Newfoundland and Labrador (Northwest Atlantic). Dots (●) represent the locations of the sampling tows.

**Figure 7 marinedrugs-18-00598-f007:**
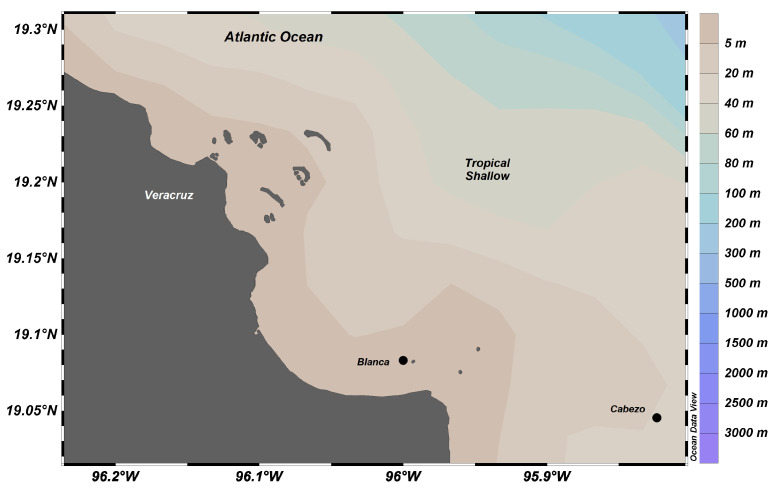
Map of sampling sites off Veracruz harbor in the coral reef system of Veracruz, Mexico. Dots (●) represent the locations of the sampling by scuba diving.

**Figure 8 marinedrugs-18-00598-f008:**
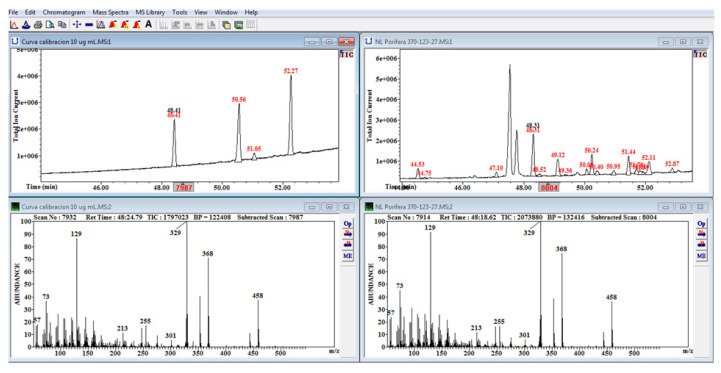
Total ion current (TIC) chromatogram and retention time (RT) of standards mix (upper left panel) including cholesterol (48.41), campesterol (50.56), stigmasterol (51.05), and β-sitosterol (52.27). The black number indicates the mass spectrum for cholesterol in the calibration curve (lower left panel). Identified chromatogram peaks of sponge *T. muricata* (upper right panel) were 24-nordehydrocholesterol (44.53), 24-nordehydrocholestanol (44.75), occelasterol (47.10), cholesterol (48.31), cholestanol (48.52), brassicasterol (49.12), brassicastanol (49.36), stellasterol (50.01), 24-methylenecholesterol (50.24), campesterol (50.40), stigmasterol (50.95), episterol (51.44), 4-24 dimethyl 5, 7-dien-3-β-ol (51.70), poriferasterol (51.81), spinasterol (51.89), β-sitosterol (52.11), and fucosterol (52.87). The black number indicates the mass spectrum for cholesterol in the sponge (lower right panel) for comparison with the standard.

**Table 1 marinedrugs-18-00598-t001:** Species systematics, sample size, water temperature, and depth of animals collected in the Northwest deep Atlantic off Newfoundland, Canada, and in the coral reef system of Veracruz, Mexico.

Scientific Name	Phyllum	Class	Order	Family	*n*	Depth (m)	Temperature (°C)
*Thenea muricata*	Porifera	Desmospongiae	Astrophorida	Pachastrellidae	4	353	4.0 ± 0.3
*Aplysina* sp.			Verongida	Aplysinidae	2	20	25.3 ± 0.9
*Pseudoanthomastus agaricus*	Cnidaria	Anthozoa	Alcyonacea	Alcyoniidae	3	1027	4.0 ± 0.3
*Montastraea cavernosa*			Scleractinia	Montastreidae	4	20	25.3 ± 0.4
*Buccinum* sp.	Mollusca	Gastropoda	Neogastropoda	Buccinidae	3	759	4.0 ± 0.3
*Pasiphaea tarda*	Arthropoda	Malacostraca	Decapoda	Pasiphaeidae	21	1321	4.0 ± 0.3
*Phormosoma placenta*	Echinodermata	Echinoidea	Echinothurioida	Phormosomatidae	3	889	4.0 ± 0.3
*Echinometra lucunter*			Echinoida	Echinometridae	4	20	25.3 ± 0.4

**Table 2 marinedrugs-18-00598-t002:** Sterol composition (percent of total sterols) and total concentration (mg·g^−1^ wet weight) of deep cold-water sponge, cnidarian, mollusk, arthropod, and echinoid representatives. Values are means ± intervals at 95% confidence.

Sterol Composition (%)	Sponge *Thenea muricata*	Cnidaria *Pseudoanthomastus agaricus*	Mollusk *Buccinum* sp.	Arthropod *Pasiphaea terda*	Echinoderm *Phormosoma placenta*	*F*	*p*
24-Nordehydrocholesterol	6.1 ± 0.8 ^a^	0.9 ± 0.7 ^b^	0.9 ± 0.1 ^b^	0.3 ± 0.3 ^b^	2.5 ± 1.3 ^b^	23.23	<0.001
24-Nordehydrocholestanol	1.3 ± 0.3	-	-	-	-		
24-Nor-22, 23 methylenecholest-5-en-3-β-ol	-	-	-	-	1.4 ± 0.3		
Occelasterol	3.2 ± 0.5 ^a^	11.6 ± 0.7 ^b^	1.9 ± 0.9 ^a^	2.2 ± 0.9 ^a^	7.2 ± 3.1 ^a,b^	18.77	<0.001
Cholesterol	22.8 ± 1.8 ^a^	68.8 ± 0.3 ^b^	91.7 ± 1.0 ^c^	95.3 ± 2.0 ^c^	73.9 ± 11.0 ^b,c^	49.82	<0.001
Cholestanol	2.0 ± 0.3 ^a^	3.6 ± 2.6 ^a^	1.5 ± 0.5 ^a^	-	1.5 ± 1.0 ^a^	2.92	0.067
9-10 Secocholesta-5(10), 6, 8 trien-3-βol (analogue of vitamin D)	-	-	-	-	1.0 ± 0.2		
Brassicasterol	12.1 ± 0.5 ^a^	8.0 ± 2.6 ^a,b^	0.8 ± 0.4 ^c^	0.7 ± 0.9 ^c^	2.9 ± 0.9 ^b,c^	57.18	<0.001
Brassicastanol	1.5 ± 0.6	-	-	-	-		
Stellasterol	3.9 ± 0.4 ^a^	-	-	-	0.3 ± 0.04 ^b^	41.13	<0.001
24-Methylenecholesterol	12.2 ± 0.4 ^a^	1.3 ± 0.3 ^b^	1.7 ± 0.8 ^b^	-	1.1 ± 0.3 ^b^	65.67	<0.001
Campesterol	2.6 ± 1.4 ^a^	2.0 ± 0.4 ^a^	-	-	2.5 ± 1.6 ^a^	0.24	0.827
Lathosterol	-	-	-	-	1.5 ± 0.9		
Stigmasterol	2.6 ± 0.3 ^a^	-	0.6 ± 0.2 ^b^	1.5 ± 0.6 ^a,b^	0.4 ± 0.1 ^b^	22.56	<0.001
Episterol	11.4 ± 1.6 ^a^	-	-	-	1.8 ± 1.4 ^b^	79.29	<0.001
4-24 Dimethyl 5, 7-dien-3-β-ol	2.9 ± 1.9	-	-	-	-		
Poriferasterol	2.0 ± 0.6	-	-	-	-		
Spinasterol	1.2 ± 0.4	-	-	-	-		
β-Sitosterol	8.9 ± 0.8 ^a^	3.8 ± 0.7 ^b^	0.9 ± 0.4 ^c^	-	2.1 ± 1.3 ^b,c^	69.47	<0.001
Fucosterol	3.5 ± 0.7	-	-	-	-		
Total sterols (GC–MS) Concentration (mg·g^−1^ wet weight) ^1^	0.6 ± 0.5 ^a^	0.59 ± 0.56 ^a^	0.5 ± 0.4 ^a^	1.6 ± 0.5 ^b^	3.8 ± 1.2 ^c^	15.39	<0.001
Total free sterols (Iatroscan) concentration (mg·g^−1^ wet weight)	0.6 ± 0.2	0.5 ± 0.2	1.4 ± 0.4	1.5 ± 1.2	0.9 ± 0.2	17.53	<0.001
Organic carbon source using sterols	Phytoplankton macroalgae, and higher plants (?)	Phytoplankton and zooplankton	Zooplankton	Zooplankton and detritus	Phytodetritus		

^1^ Quantification of sterols (μg·mL^−1^ of hexane) of samples was achieved by dividing the areas from integrated chromatographic peaks by their respective slope obtained with the calibration curve [29] (pp. 18–19). Sum of sterol concentration (µg·mL^−1^) was normalized to wet tissue biomass, multiplying the concentration by hexane volume (mL) added to each sample and dividing by lipid extracted wet biomass (mg), to obtain total sterols expressed as µg·mg^−1^, equivalent to mg·g^−1^. To obtain mg per 100 g, it was multiplied by 100. ^a–c^ Different superscript letters denote significant differences among columns (*p* < 0.001).

**Table 3 marinedrugs-18-00598-t003:** Wet weight (WW) per organism, total lipids, stable isotopes, and C:N molar ratio of animals collected in the deep Northwest Atlantic off Newfoundland. Values are mean ± 95% confidence interval. Information obtained from [28]. The sponge *Aplysina* sp., coral *M. cavernosa*, and sea urchin *E. lucunter* were collected in the shallow, tropical coral reef system of Veracruz [31]. TAG, triacylglycerol; ST, sterol.

Scientific Name	Wet Weight (g)	Lipid Content (mg·g^−1^ WW)	TAG:ST Ratio	δ^13^C	Organic Carbon Source	δ15N	C:N Molar Ratio	Functional Group
*Thenea muricata*	16.2 ± 2.1 ^b^	2.6 ± 1.2 ^a^	0.3	−17.4 ± 0.4 ^b^	Pelagic	14.2 ± 0.3 ^a^	5.4 ± 0.3 ^b^	Filter feeder zooplanktivore
*Pseudoanthomastus agaricus*	12.2 ± 8.0 ^a,b^	4.1 ± 2.2 ^a^	0.4	−13.8 ± 4.9 ^b^	Bentho- pelagic	11.0 ± 1.4 ^b,c^	6.5 ± 3.1 ^a,b,c^	Omnivore detritivore
*Buccinum* sp.	5.8 ± 2.9 ^c^	6.9 ± 0.3 ^b^	0.02	−17.0 ± 1.0 ^b^	Pelagic	12.6 ± 0.6 ^b^	4.2 ± 0.1 ^a^	Carnivore scavenger
*Pasiphaea tarda*	29.2 ± 6.6 ^a^	8.7 ± 2.3 ^b^	0.5	−19.2 ± 0.1 ^a^	Pelagic	11.4 ± 0.1 ^c^	3.8 ± 0.04 ^a^	Zooplanktivore
*Phormosoma placenta*	19.6 ± 8.4 ^a,b^	6.0 ± 3.1 ^a,b^	0.5	−14.3 ± 0.9 ^c^	Bentho- pelagic	12.3 ± 0.3 ^b^	5.3 ± 0.5 ^b^	Omnivore carnivore
*Aplysina* sp.	28 ± 11 ^a^	15.4 ± 5.4 ^b,c^	0.4 ± 0.1		Pelagic			Filter feeder
*Montastraea cavernosa*	9.8 ± 3.8 ^c^	7.2 ± 3.6 ^b,c^	1.4 ± 0.5	−11.1 ± 2.5 ^c,d^	Symbiont	4.0 ± 1.9 ^d^	16.9 ± 9.5 ^c,1^	Zooxanthellae and zooplankton
*Echinometra. lucunter*	6.9 ± 1.6 ^c^	12.6 ± 1.2 ^c^	25 ± 14	−10 ± 0.1 ^d^	Macroalgae	3.3 ± 0.3 ^d^	8.5 ± 0.4 ^c^	Herbivore

^1^ The high variation was probably due to carbonate residuals. ^a–d^ Different superscript letters denote significant differences among columns (*p* < 0.05).

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
