# Peer review of "Sterol Composition of Sponges, Cnidarians, Arthropods, Mollusks, and Echinoderms from the Deep Northwest Atlantic: A Comparison with Shallow Coastal Gulf of Mexico"

_marinedrugs, 2020, doi:10.3390/md18120598_

Round 1

Reviewer 1 Report

This manuscript describes the sterol profiling of marine organisms by gas chromatography – mass spectrometry, which have different sterol synthesis pathways but, due to differences in habitat (deep-see versus shallow tropical waters) and dietary sterol intake as a result of their position in the food chain, show – at least in part – different overall sterol composition. Additional profiling information on the marine organisms were provided in form of wet weight, total lipid content, stable isotope content and C:N mol ratio.

The topic of the manuscript is highly relevant to field of natural product analysis in marine organisms. This pilot-study is innovative and appears well conducted. The manuscript is clearly structured and the conclusions are supported by the data. However, the Materials and Method section requires attention.

Thus, before a publication in Marine Drugs can be considered, this manuscript would require revision considering the 5 comments listed below.

Comment 1: The keywords should contain “gas chromatography” and “mass spectrometry”.

Comment 2: On Page 6, Line 269 the authors describe the GC-MS instrumentation as “gas chromatograph coupled to a mass spectrometer (HP 6890)”, but 6890 just describes the GC mainframe module without mentioning the detector module (e.g. FID). The type, model and manufacturer of the mass spectrometer including type of ion source needs to be mentioned as well.

Comment 3: In the material and methods section on Page 6, Line 184 please provide information on the ionisation mode, positive or negative, that was used for the mass spectrometry.

Comment 4: In the material and methods section on Page 6, Line 185 please provide information on the reference data bank that was used for sterol identification, if applicable.

Comment 5: The manuscript requires some corrections of the scientific terms and English language e.g.:

Page 3, Line 113: Replace “Philogenic” with “Phylogenetic”.

Page 3, Line 113: Replace “Cnidarian” with “Cnidaria”.

Page 4, Lines 119-120: Replace “The geographical comparison of the northwest Atlantic and southwest Gulf of Mexico is shown in Figure 2” with “The geographical location and depth-profile of the northwest Atlantic and southwest Gulf of Mexico sites are shown for comparison in Figures 2 and 3”.

Page 4, Line 120: Explain “zone 3K”.

Page 6, Line 163: Replace “was” with “were”.

Page 7, Line 213: In “it was the only phylum” it is unclear to what phylum “it” is referring to.

Page 11, Line 303: Replace “And” with “and”.

Page 12, Line 346: Replace “dealkilation” with “dealkylation”.

Page 13, Line 376: Replace “Depratment” with “Department”.

According to IUPAC nomenclature the abbreviation of liter is “L” not ”l”. Please correct throughout the manuscript.

References: Please review capitalization of title words.

Reference 17: Italicize “Placopecten magellanicus”.

Reference 42: Replace “In” with “In:”.

Author Response

Comments from reviewer #1

This manuscript describes the sterol profiling of marine organisms by gas chromatography – mass spectrometry, which have different sterol synthesis pathways but, due to differences in habitat (deep-see versus shallow tropical waters) and dietary sterol intake as a result of their position in the food chain, show – at least in part – different overall sterol composition. Additional profiling information on the marine organisms were provided in form of wet weight, total lipid content, stable isotope content and C:N mol ratio.

The topic of the manuscript is highly relevant to field of natural product analysis in marine organisms. This pilot-study is innovative and appears well conducted. The manuscript is clearly structured and the conclusions are supported by the data. However, the Materials and Method section requires attention.

Thus, before a publication in Marine Drugs can be considered, this manuscript would require revision considering the 5 comments listed below.

We would like to thank reviewer #1 for their comments

Comment 1: The keywords should contain “gas chromatography” and “mass spectrometry”.

We added the key words “gas chromatography” and “mass spectrometry” on lines 36 and 37

Comment 2: On Page 6, Line 269 the authors describe the GC-MS instrumentation as “gas chromatograph coupled to a mass spectrometer (HP 6890)”, but 6890 just describes the GC mainframe module without mentioning the detector module (e.g. FID). The type, model and manufacturer of the mass spectrometer including type of ion source needs to be mentioned as well.

We added the description of MS coupled to HP 6890 on line 437 as follows: GC-MS (EI): Agilent 6890N GC - 5973 MSD (quadrupole) with…

Comment 3: In the material and methods section on Page 6, Line 184 please provide information on the ionisation mode, positive or negative, that was used for the mass spectrometry.

We added the ionisation mode in line 440-441 as follows: The positive ionization voltage on the repeller pushes the positive ions into several electrostatic lenses producing a positive ionization mode.

Comment 4: In the material and methods section on Page 6, Line 185 please provide information on the reference data bank that was used for sterol identification, if applicable.

We provided information of identification process in lines 443-455.

Positions of peaks were useful for identification of sterols with the same molecular ion and the list provided by [20] was used to infer position relative to cholesterol. Mass spectra were searched with Wsearch 32 software (Wsearch 2008; version 1.6 2005) and the NIST Mass Spectral Search Program for the NIST/EPA/NIH Mass Spectral Library (Demo version 2.0 f. build Apr 1 2009). Sterol identity was confirmed by mass spectra interpretation using the molecular weight (MW) ion which provides information on number of carbons and double bonds [31, 47]. Some important fragments for interpretation of the whole structure are (M-R)+, (M-R-2H)+, (M-H2O)+, (MR-H2O)+, (M-R-2H-H2O)+, and (M-CH3-H2O)+. Other fragments indicate the position of double bonds. For instance, lathosterol-TMS ether has a MW=458. This means the molecule has 27 carbons and one double bond. We should expect a signal at m/z=213 indicating a double bond at C7. For stigmasterol-TMS ether, double bonds at C5 and C22 are shown by ions with m/z=129 and m/z=255 [31] (p. 15), [47] (p. 235). Finally, three standards were injected to build a calibration curves for cholesterol, campesterol and β-sitosterol.

Comment 5: The manuscript requires some corrections of the scientific terms and English language e.g.:

Page 3, Line 113: Replace “Philogenic” with “Phylogenetic”.

We replaced “Philogenic” with “Phylogenetic”

Page 3, Line 113: Replace “Cnidarian” with “Cnidaria”.

According to the Merrian-Webster dictionary cnidarian (singular) and cnidarians (plural) can be used as a noun; however the reviewer is right about the formal term of the phylum which is Cnidaria. We changed from trivial to formal names in line 125-126 as follows: To test our hypothesis, we compared sterol profiles of Porifera, Cnidaria, and Echinodermata from two contrasting sites…, However,  other parts of the text we mention trivial names, such as sponges, cnidarians and echinoderms.

Page 4, Lines 119-120: Replace “The geographical comparison of the northwest Atlantic and southwest Gulf of Mexico is shown in Figure 2” with “The geographical location and depth-profile of the northwest Atlantic and southwest Gulf of Mexico sites are shown for comparison in Figures 2 and 3”.

We replaced the sentence in lines 357-358 as follows: The geographical location and depth-profile of the northwest Atlantic and southwest Gulf of Mexico sites are shown for comparison in Figures 5 and 6. The Figure numbers are changed following the Marine Drugs template as suggested by reviewer #2 and the editor.

Page 4, Line 120: Explain “zone 3K”.

We added the explanation in lines 360-362 as follows: Zone 3K represents the marine area located off the northern coasts of Newfoundland, according to the Northwest Atlantic Fisheries Organization NAFO division of the North Atlantic Ocean (Fig. 6; more details in [48]).

We added the reference 48. https://www.nafo.int/About-us/Maps

Page 6, Line 163: Replace “was” with “were”.

We replaced “was” with “were” in line 407 as follows: Sponges, cnidarians and sea echinoderms sterols from the deep, cold were compared to…

Page 7, Line 213: In “it was the only phylum” it is unclear to what phylum “it” is referring to.

We made it clear to which phylum “it” is referring to in line 141 as follows: Furthermore, from the deep, cold site Porifera was the only phylum showing…

Page 11, Line 303: Replace “And” with “and”.

We replaced “And” with “and” in line 292 as follows: The mollusk Buccinum sp. and the arthropod P. tarda

Page 12, Line 346: Replace “dealkilation” with “dealkylation”.

We replaced “dealkilation” with “dealkylation” in line 343 as follows: echinoderms can modify phytosterols by dealkylation or synthesize cholesterol…

Page 13, Line 376: Replace “Depratment” with “Department”.

We replaced “Depratment” with “Department” in line 518 as follows: Acknowledgments: We thank D. Stansbury, K. Tipple, D. Pittman, V.E. Wareham, from Department…

According to IUPAC nomenclature the abbreviation of liter is “L” not ”l”. Please correct throughout the manuscript.

We replaced ”l” with “L” in line 416 as follows: Samples were weighed wet and around 0.7±0.2 g was immersed in 4 mL of chloroform.

References: Please review capitalization of title words.

We reviewed capitalization on title words of all references following template instructions

Reference 17: Italicize “Placopecten magellanicus”.

We italicized Placopecten magellanicus in line 561

Reference 42: Replace “In” with “In:”.

We replaced “In” with “In:” in line 621

Reviewer 2 Report

The manuscript by Laura Carreón-Palau describe Sterol composition of sponges, cnidarians, arthropods, mollusks, and echinoderms from the deep Northwest Atlantic. After close evaluation of paper I recommend revision according to the next points:

  1. Please follow the template recommended by Marine drugs. The section 2. Materials and Methods should be located after Discussion.
  2. Fig. 2 is appears in line 127, but refered in line 166 jnly. While Fig.3 is refered before Fig.2. Please correct.
  3. In phrase "Mean bottom temperature at the sampling site ranged between 3.2 and 4.5°C..." please clarify at which month (or this is mean year temp).
  4. Line 124- what was a reason to refer 2012? Are there more bew data from 2019-2020?
  5. Please check the correct name for Phoromosa placenta (Table 1)
  6. In line 168 - "Samples were weighed wet and around 0.7 ± 0.2 g..." Which samples you mind? The wet weight in Table 2 are much higher.
  7. Please describe in details procedure for lipids isolation and identification.
  8. Line 185 - please indicate manufacturers for references, which references were used?
  9. Please provide the typical TIC chromatogram and mark the peaks which were identified. The chromatogram of references for comparison is required also.
  10. In Table 2 please clarify small letters a,b,c. What is mind "-"?
  11. The title of Table 3 indicate "...(percent of total sterols) and total concentration (mg g-1 wet wt)". How to distinguish in the table percent and concentration?
  12. Line 247 - check units"5.4% mg g-1"
  13. Fig. 6 - statistical deviations are required. For some spp. the number of samples was not enough (2-3 see Table 1). Please indicate statistically significant differences and discuss this aspect.
  14. In Discussion I would suggest to compare results about sterols composition with other published papers (see https://doi.org/10.3390/molecules25184088; https://doi.org/10.1016/B978-0-12-819570-3.00004-4; https://doi.org/10.1007/s00227-019-3544-y; https://doi.org/10.1016/j.jff.2015.05.030; https://doi.org/10.3390/md15120365; https://doi.org/10.1016/j.steroids.2018.07.003; )

Author Response

Comments of reviewer #2

The manuscript by Laura Carreón-Palau describes Sterol composition of sponges, cnidarians, arthropods, mollusks, and echinoderms from the deep Northwest Atlantic. After close evaluation of paper I recommend revision according to the next points:

  1. Please follow the template recommended by Marine drugs. The section 2. Materials and Methods should be located after Discussion.

We followed the template recommended by Marine Drugs. We moved section 2 after the discussion in lines 356 to 488.

  1. 2 is appears in line 127, but referred in line 166 jnly. While Fig.3 is referred before Fig.2. Please correct.

We corrected the order of Figures according to change of section. Figures 2, 3 and 4 changed to Figures 5, 6 and 7 while Figures 5, 6 and 7 changed to Figures 2, 3 and 4.

  1. In phrase "Mean bottom temperature at the sampling site ranged between 3.2 and 4.5°C..." please clarify at which month (or this is mean year temp).

These numbers are on a daily basis. Sampling was carried out in the months of November and December 2013, we added this information on lines 363-364 as follows: Mean daily bottom temperature during sampling period at the sampling site ranged between 3.2 and 4.5°C, with a slight decrease with depth.

Line 124- what was a reason to refer 2012? Are there more bew data from 2019-2020?

Ideally, we should get chlorophyll concentrations from the same month and year of sampling. Unfortunately, we did not find information regarding November-December 2013 and the closest year was 2012.

Please check the correct name for Phoromosa placenta (Table 1)

Many thanks for pointing out this typo. We have now corrected the name and replaced it with “Phormosoma placenta” in line 38 as follows: ...Buccinum sp., Pasiphaea tarda, Phormosoma  placenta. Also in line 149 when species was mentioned by the first time as follows: ...The echinoderm  Phormosoma placenta had…and in Tables 1,2, and 3.

  1. In line 168 - "Samples were weighed wet and around 0.7 ± 0.2 g..." Which samples you mind? The wet weights in Table 2 are much higher.

Table 3 provide wet weights of whole-bodied organisms, whereas lines 400 provide the amounts of tissues collected for analysis. We have now modified lines 399-402 as follows: A total of 34 deep-sea organisms, belonging to 5 species from different phyla, were weighed, and samples of 0.7±0.2 g wet weight were collected from each individual and processed for lipid analysis at at the Core Research Equipment and Instrument Training-Aquatic Research cluster (CREAIT-ARC) Facility at Memorial University.

  1. Please describe in details procedure for lipids isolation and identification.

We added the details procedure for lipid isolation and identification on lines 417-430 as follows: Lipids were extracted adding 2 mL of methanol and 1.8 mL of distilled water, mixed with a vortex and centrifuged to separate polar and non-polar phases. The lower phase was recovered in a lipid clean tube; an aliquot was separated into lipid classes by Chromarod thin-layer chromatography (TLC) (Parrish, 1987a). Chromarod TLC and Iatroscan flame ionization detection (FID) were used to quantify the lipid classes in the extracts. TLC/FID was performed with Chromarods SIII and an Iatroscan MK VI (Iatron Laboratories, Tokyo). Concentrations of triacylglicerols (TAG) and steryl esters and free sterols were obtained by interpolation with a calibration curve constructed with five amounts, ranging between 0.5 and 4.0 μg, of the following standards: cholesteryl stearate, triplamitin, and cholesterol (Sigma-Aldrich). The rest of the extract was separated by solid phase extraction column (Strata SI-1 Silica 55 mm, 70 A; Phenomenex). Neutral lipid fractions were recovered with 6 mL of chloroform: methanol: formic acid (98 : 1 : 0.05), then galactoglycerolipids were recovered with 6 mL of 100% acetone and phosphoglycerolipids were recovered with 9 mL of chloroform: methanol (5 : 4) as a modification of [45].

  1. Line 185 - please indicate manufacturers for references, which references were used?

This is a misunderstanding. To clarify we replaced "The references for sterol identification were [20] and [47] (p. 235)" with "Sterol were identified by reference to [20] and [47] (p. 235). A detailed explanation of identification was added as Reviewer #1 and #2 suggested in lines 442 to 455 as follows:

Sterols were identified by reference to [20] and [47] (p. 235). Positions of peaks were useful for identification of sterols with the same molecular ion and the list provided by [20] was used to infer position related to cholesterol. Mass spectra were searched with Wsearch 32 software (Wsearch 2008; version 1.6 2005) and the NIST Mass Spectral Search Program for the NIST/EPA/NIH Mass Spectral Library (Demo version 2.0 f. build Apr 1 2009). Sterol identity was confirmed by mass spectra interpretation using the molecular weight (MW) ion (M+) which provides information of number of carbons and double bonds [31, 47]. Some important fragments for interpretation of the whole structure were (M-R)+, (M-R-2H)+, (M-H2O)+, (MR-H2O)+, (M-R-2H-H2O)+, and (M-CH3-H2O)+. Other fragments indicate the position of double bonds. For instance, lathosterol-TMS ether will have a MW=458. This means the molecule has 27 carbons and one double bond. We should expect a signal at m/z=213 indicating a double bond at C7. For stigmasterol-TMS ether, double bonds at C5 and C22 are shown by ions with m/z=129 and m/z=255 [31] (p. 15), [47] (p. 235). Finally, three standards were injected to build a calibration curves for cholesterol, campesterol and β-sitosterol (Sigma-Aldrich).

  1. Please provide the typical TIC chromatogram and mark the peaks which were identified. The chromatogram of references for comparison is required also.

We added Figure 8. Total Ion Current (TIC) chromatogram and retention time (RT) of standards mix (upper left panel) including cholesterol (48.41), campesterol (50.56), stigmasterol (51.05) and β-sitosterol (52.27), black number indicates mass spectrum for cholesterol in the calibration curve (lower left panel).  Chromatogram of sponge T. muricata (upper right panel) peaks identified were 24-nordehydrocholesterol (44.53), 24-nordehydrocholestanol (44.75), occelasterol (47.10), cholesterol (48.31), cholestanol (48.52), brassicasterol (49.12), brassicastanol (49.36), stellasterol (50.01), 24-methylenecholesterol (50.24), campesterol (50.40), stigmasterol (50.95), episterol (51.44), 4-24 dimethyl 5, 7-dien-3-β-ol (51.70), poriferasterol (51.81), spinasterol (51.89), β-sitosterol (52.11), fucosterol (52.87). Black number indicates mass spectrum for cholesterol in the sponge (lower right panel) for comparison with standard.

  1. In Table 2 please clarify small letters a,b,c. What is mind "-"?

We added the following sentence to the title of Table 2: Letters denote significant differences among columns.

  1. The title of Table 3 indicate "...(percent of total sterols) and total concentration (mg g-1 wet wt)". How to distinguish in the table percent and concentration?

We added the sentence: Sterol composition (%) to the column title and the phrase total sterol concentration (mg g-1 wet wt) to the respective row. Concentration refers to the sum of sterols, while the proportion of each sterol is related to the total concentration. Therefore, if the reader needs to know the concentration of some sterol, it can be calculated, for instance, cholesterol of T. muricata in Table 2 is 22.8% of 0.6 mg g-1 equivalent to 0.14 mg g-1 WW.  

  1. Line 247 - check units"5.4% mg g-1"

We revised the units and removed % as we are talking about concentration in line 175 as follows: had 15.4±5.4 mg g-1 wet weight…

  1. 6 - statistical deviations are required. For some spp. the number of samples was not enough (2-3 see Table 1). Please indicate statistically significant differences and discuss this aspect.

We added error bars denoting 95% of confidence interval in Figures 3 and 4. As we said in comment 2 of reviewer #2 in the previous version were Figs. 6 and 7. Also, we added a paragraph in the discussion section in lines 306 to 314 as follows: Small sample sizes were not a statistical issue due to small variation and clear differences among species and sites. Permutational multivariate analysis of variance (PERMANOVA), allows multivariate comparisons of compositional data such as sterol profiles with data with or without a normal distribution of residuals. Here, 996 permutations were performed in the main test, and in those species with small sample size such as sponges, 10 permutations allowed detecting significant differences (p=0.001). Also, between cnidarians there were 56 permutations allowing detection of significant differences (p=0.001). In contrast, echinoderms with 35 permutations did not show significant difference (p=0.082) because sterol profiles of echinoderms from shallow and deep sites were quite similar.  

  1. In Discussion I would suggest to compare results about sterols composition with other published papers (see https://doi.org/10.3390/molecules25184088; https://doi.org/10.1016/B978-0-12-819570-3.00004-4; https://doi.org/10.1007/s00227-019-3544-y; https://doi.org/10.1016/j.jff.2015.05.030; https://doi.org/10.3390/md15120365; https://doi.org/10.1016/j.steroids.2018.07.003; https://doi.org/10.1073/pnas.1512614113 )

We read the papers suggested and we added the reference https://doi.org/10.1073/pnas.1512614113 [50] in line 331 of the discussion. The rest of the papers were cited in the conclusion because they are more related with bioactive metabolites and their function against diabetes and inflammatory diseases that we did not investigate. However, future research can include detection of those compounds after modifying the analytical methods. We modified the conclusion on lines 491-507 as follows:

In the deep cold sites, gastropods and arthropods had cholesterol as the main sterol. Other extraction systems such as acetic acid and methanol could facilitate finding bioactive sterol metabolites for use against diabetes and oxidative stress-induced inflammatory diseases [49]. Sponges and cnidarians from shallow, tropical environments had plant and zooxanthellae sterols, respectively, while cold, deep sponges had 24-ethyl sterols probably synthesized by themselves [50], and cnidarians had phytoplankton, zooplankton, and probably fungal sterols. In contrast, echinoderms, the most complex phylum along with hemichordates and chordates (deuterostomes), did not show significant differences in their sterol profiles suggesting that cholesterol synthesis is present in all deuterostomes. Tropical, shallow sponges, cnidarians and echinoderms showed a significantly higher concentration of lipids, as well as higher TAG:ST ratios, suggesting a better nutritional condition than deep, cold invertebrates. However; other echinoderm groups should be studied such as holothurians which did not have cholesterol as the main sterol, but rather diatomsterol, probably because supercritical fluid extraction was used [51]. Sex was the main factor affecting sterol amount in the gonads (intact, spawned) and gametes of echinoderm A. dufresnii, with differences in their total concentration from intact to spawn explained by gamete concentrations [52]. As gonads are a good functional food for diabetes mellitus treatment in mice [53], sex and gonad composition of echinoderms should be recorded in future surveys.

Round 2

Reviewer 2 Report

Authors have revised the manuscript, however, some points needs additional attention

  1. Authors still not followed to the template of journal. (see for example subsections numbering sect. 3 and 4)
  2. I would suggest to discuss https://doi.org/10.3390/md15120365 which was recommended in the first round. Similar setrols were identified in body wall of sea urchins endemic to cold waters.
  3. The title of Table 2 should be modified. Please indicate in the title the method, which was used for quantification of total concentration (mg g-1 wet wt). Authors have not used references for GC-MS, how quantification was done?
